# The Role of MRI in the Diagnosis of Solid Pseudopapillary Neoplasm of the Pancreas and Its Mimickers: A Case-Based Review with Emphasis on Differential Diagnosis

**DOI:** 10.3390/diagnostics13061074

**Published:** 2023-03-13

**Authors:** Jelena Djokic Kovac, Aleksandra Djikic-Rom, Aleksandar Bogdanovic, Aleksandra Jankovic, Nikica Grubor, Goran Djuricic, Vladimir Dugalic

**Affiliations:** 1Center for Radiology and Magnetic Resonance Imaging, University Clinical Centre of Serbia, Pasterova No. 2, 11000 Belgrade, Serbia; 2Faculty of Medicine, University of Belgrade, Dr Subotica No. 8, 11000 Belgrade, Serbia; 3Departament of Pathology, Clinic for Digestive Surgery, University Clinical Centre of Serbia, Pasterova No. 2, 11000 Belgrade, Serbia; 4Clinic for Digestive Surgery, University Clinical Centre of Serbia, Koste Todorovica Street, No. 6, 11000 Belgrade, Serbia; 5Departament for Radiology, University Children’s Clinic, Tiršova Street, No. 10, 11000 Belgrade, Serbia

**Keywords:** solid pseudopapillary neoplasm, mimickers, magnetic resonance imaging

## Abstract

Solid pseudopapillary neoplasm (SPN) is rare pancreatic tumor occurring most commonly in young females. The typical imaging appearance of SPN is of well-defined, encapsulated, and large heterogeneous tumors, consisting of solid and cystic components due to various degrees of intralesional hemorrhage and necrosis. However, atypical imaging presentation in the form of small solid tumors or uniformly cystic lesions might also be seen, which can be explained by specific pathological characteristics. Other imaging features such as a round shape, the absence of main pancreatic duct dilatation, and slow growth, in combination with vague symptoms, favor the diagnosis of SPNs. Nevertheless, the radiological findings of SPN might overlap with other solid and cystic pancreatic neoplasms, such as neuroendocrine tumors, serous and mucinous neoplasms, and even small pancreatic adenocarcinomas. In addition, a few benign non-tumorous conditions including walled-of-necrosis, and intrapancreatic accessory spleen may also pose diagnostic dilemmas simulating SPNs on imaging studies. The aim of this manuscript is to provide a comprehensive overview of the typical and atypical imaging features of SPNs and to describe useful tips for differential diagnosis with its potential mimickers.

## 1. Introduction

Solid pseudopapillary neoplasm (SPN) is a rare pancreatic tumor comprising 2% of all exocrine pancreatic neoplasms, and 9.3% of all cystic pancreatic tumors [1,2]. These uncommon neoplasms were first described by Frantz in 1959 as solid and papillar pancreatic tumors. Since then, various terms have been used for these lesions, until WHO classified them as a solid pseudopapillary neoplasms in 1996 [3,4]. SPNs occur predominantly in women younger than 40 years [1]. Although rarely, SPNs have also been described in males, and in comparison to females, they usually occur in an older age group and display more aggressive behavior [5].

Typically, these tumors present with vague, non-specific symptoms such as abdominal discomfort, pain, or palpable upper abdominal mass [1]. Nausea and vomiting were also reported [6,7]. Jaundice is rarely present [1]. Due to the slow growth, SPNs are asymptomatic in approximately 15% of patients, and they are frequently incidentally detected [1,7].

From a histological point of view, these are the tumors of epithelial origin, but their histogenesis is still not elucidated. The pathogenesis of SPNs is unclear because the origin of cells does not resemble any cell type in the embryonic or adult pancreas [2,6]. 

Since the incidence is highest in younger women, a close relation between the tumor and female sex hormones might be suggested, but it is still not confirmed [8,9]. Due to the female predilection, it has been hypothesized that SPNs originate from genital ridges close to the pancreatic anlage during organogenesis [10]. According to some other studies, SPNs arise from pluripotential embryonic stem cells [9]. Pathologically, these enigmatic tumors arise as solid lesions with subsequent cystic degeneration as the tumor outgrows the inadequate blood supply. Degenerative changes lead to loss of neoplastic tissue with formation of pseudopapillae and development of heterogeneous lesions consisting of variable proportion of solid, cystic, and hemorrhagic components [9].

The most common imaging appearance of SPNs is of well-defined, encapsulated, and large heterogeneous tumors, consisting of solid and cystic components due to various degrees of intralesional hemorrhage and necrosis [11]. Although the typical imaging features of SPNs have been described in many previous reports, an accurate preoperative diagnosis of this rare entity remains low. One of the reasons for this is that the atypical forms of SPN have a more difficult differentiation from other, more common pancreatic lesions [12]. In addition to atypical forms of SPNs, radiological features of typical SPNs might overlap with those of more common benign and malignant pancreatic lesions [13]. The widespread use of cross-sectional imaging modalities, in particular computed tomography (CT) and magnetic resonance imaging (MRI), led to an increased detection of smaller SPNs, which commonly appear as solid lesions. In clinical practice, these solid SPNs are frequently misdiagnosed as pancreatic ductal adenocarcinomas (PDACs) or neuroendocrine tumors (NET) [14]. Furthermore, if SPN develops in patients where we epidemiologically do not expect these lesions, such as in males or older females, diagnostic errors might occur. Considering high soft tissue resolution, MRI is the preferred imaging modality for the characterization of cystic pancreatic tumors and complex tumors composed of both solid and cystic components, such as SPNs. Nevertheless, precise differentiation is not always possible and further evaluation with endoscopic ultrasound (EUS) is recommended [15]. EUS provides a detailed assessment of the cystic morphology, and moreover, enables fine-needle aspiration and the subsequent analysis of the cyst content. At the moment, the analysis of cyst fluid cytology and cyst fluid tumor markers, such as carcinoembryonic antigen, has been commonly used to improve the distinction between mucinous and non-mucinous cysts [15].

Taking into account the rarity and variability of SPN imaging presentation, the precise preoperative diagnosis presents a real diagnostic challenge, even for experienced radiologists. Since SPNs are tumors with indolent biological behaviors, excellent prognosis after complete tumor resection, and a cure rate up to 90% [16,17]; accurate preoperative distinction from other more aggressive tumors is clinically very important. Therefore, the aim of this study is to provide a detailed review of the typical and atypical imaging features of SPNs, and also to show benign and malignant pancreatic lesions that can resemble SPNs. In addition, useful tips for the differentiation of SPN from its potential mimickers are highlighted.

## 2. Typical Imaging Presentation of Solid Pseudopapillary Neoplasms

SPNs are most commonly large, round, well-defined, and encapsulated heterogeneous tumors [18]. Typically, SPNs consist of both cystic and solid components, which are present in variable proportions. Thus, some of the lesions might appear predominantly as a solid with small intralesional cystic areas, while the other lesions are mainly cystic with solid parts located peripherally [19]. Although SPNs can occur in all parts of the pancreas, there is a slight predilection for the pancreatic tail [6]. The wall of the lesions is smooth in the majority of cases, but it can also be nodular [20]. On T1-weighted images, the internal content is hypointense, but usually with hyperintense areas due to internal hemorrhage [20]. On T2-weighted images, the most common appearance is heterogeneously hyperintense. Typically, a solid component enhances poorly in comparison to the rest of the pancreas in the arterial phase with a progressive enhancement in the portal-venous and delayed phase [11] (Figure 1). 

While larger lesions display heterogeneous enhancement, smaller tumors enhance homogeneously [11]. Internal hemorrhage is considered to be pathognomonic finding for these rare neoplasms, as was reported in 29% up to 88.9% of cases [20,21] (Figure 2). The internal fluid–fluid levels might also be seen [22]. A peripheral capsule is almost always present in tumors that are larger than 3 cm. On an MRI, the capsule is typically hypointense on both T1- and T2-weighted sequences with a moderate enhancement after intravenous contrast administration [21]. Calcifications are seen in up to 30% of SPNs [2,23,24], and are more frequently encountered in larger tumors [6,11]. A variety of calcification patterns may occur, ranging from peripheral to central, and being coarse or faintly amorphous [14,25].

Imaging presentations of SPNs correspond well to their histopathological composition [26]. On gross specimens, these tumors are soft, which can explain the rarity of main pancreatic duct dilatation [23]. Additionally, even when they are located in the head of the pancreas, jaundice is rarely present [24]. Due to the fibrous pseudocapsule, these tumors are well-defined [23]. Histologically, SPNs consist of delicate papillary fronds and uniform epithelioid cells, which are arranged in nests [27]. Tumor cell decohesion occurs due to the degenerative changes, while typical pseudopapillary structures consist of fibrovascular stalks covered by sheets of viable tumor cells [2,24,27]. The internal hemorrhage, which is typically seen in the majority of SPNs, might be explained by insufficient blood supply in large lesions [27].

SPNs were classified as lesions with uncertain malignant potential in the latest WHO classification [2]. Although the majority of tumors display benign behavior, malignancy has been reported in 10–15% of cases [2,28,29]. On histology, perineural invasion, angioinvasion, cellular and nuclear atypia, high mitotic rate, and extensive necrosis are reported to indicate the malignant potential of SPNs [2,30]. There are a few reports describing imaging features, which imply the malignant potential of SPNs [31,32]. In this setting, Lee et al. in their study showed that pancreatic duct dilatation, vessel encasement, and the presence of metastases were the only significant predictors of malignant SPNs [19]. On the other hand, there was no difference in tumor size, location, capsule thickness, internal content, and calcification pattern between benign and malignant SPNs [19]. Distant metastases are present in up to 15% of patients with SPNs, and are usually already seen at the time of diagnosis [29]. However, metastases may develop years after surgery, which indicates the importance of long-term follow-ups, even in the tumors considered as low-grade malignancy [33]. The most common sites for metastases are the liver, peritoneum, and lymph nodes [29,34,35,36]. Liver metastases may be multiple, but they are usually solitary [31]. Nevertheless, even in the case of metastatic SPNs, long-term survival has been described due to the slow-growing nature of the tumor [37]. Although angioinvasion has been described as one of pathohistological findings in malignant SPNs, vessel encasement is rarely seen preoperatively [2]. Additional imaging features that are indicative for malignancy are focal capsular discontinuity and the presence of lobulated margins [31,38,39]. Even though the preoperative recognition of malignant SPNs is important from a surgical perspective, it is quite difficult when based on the imaging criteria only [40]. Malignant behavior might be suspected only in cases of large tumors with vessel encasement, main pancreatic duct dilatation, or in the case of metastatic disease [31]. Concerning the tumor size as a predictive factor for malignancy, there are conflicting results in the present literature. While in the study by De Robertis et al., no significant difference was found between small and large SPNs, the size of the lesion was found to be one of the predictive factors for aggressiveness and recurrence after surgery in other reports [6,31,41,42]. Nevertheless, it has been reported that metastases can occur even in the absence of histological confirmation of malignancy in resected lesions [3]. Therefore, SPNs are classified as lesions of uncertain malignant potential [3].

## 3. Solid Pseudopapillary Neoplasm Mimicking Other Solid Pancreatic Lesions

Although cystic degeneration is typical for SPNs, small tumors less than 3 cm in diameter may present as entirely solid lesions [25,39]. It is hypothesized that all SPNs are solid lesions when they are small in diameter. With the growth of the lesion, the blood supply becomes insufficient, which leads to cystic degeneration and the development of internal hemorrhage [7]. Since the correct distinction between small solid SPNs and other solid pancreatic tumors with higher malignant potential would help in the selection of the appropriate treatment, it is important to highlight the imaging features of these uncommon lesions. On T1-weighted images, small solid SPNs appear as hypointense lesions, while on T2-weighted images, these lesions typically have a high or very high signal intensity [43] (Figure 3). This finding can be explained by the composition of SPNs, which consist of solid sheets and nests of cells with an abundant cytoplasm, pseudopapillary formation, and variable degenerative changes [27,43]. Additionally, small SPNs are less sharply circumscribed, and they often lack peripheral capsules [6]. Regarding the shape of the lesion, small solid SPNs are mostly round [39]. If a lobulated shape is present, it could indicate a malignant potential, according to some studies [19,31]. Similarly to larger SPNs, in small tumors, early peripheral heterogeneous enhancement with progressive fill-in on contrast-enhanced MRIs has been described as typical vascular behavior [5]. Nevertheless, it should be kept in mind that the enhancement of small tumors is weaker in comparison to larger ones. This finding might further complicate reaching an accurate preoperative diagnosis. 

Rarely, large SPNs might also display a completely solid pattern [6] (Figure 4). 

Due to the lack of characteristic imaging features, differential diagnosis between solid SPNs and other solid pancreatic lesions might be very difficult [44]. In the study by Chae et al., a correct preoperative diagnosis was made in only six out of eleven small SPNs [21]. The difficulties in preoperative diagnosis of small SPNs have also been pointed out in other reports [5,6]. Thus, the rate of incorrect diagnosis was near 40% in the study by De Robertis et al. [6]. The most common misdiagnoses in previous studies were NETs, followed by PDACs [39,43,45]. In order to correctly differentiate SPNs from other solid tumors with a higher malignant potential or from completely benign lesions, following imaging features must be considered: enhancement pattern, margins, and secondary signs such as the presence of main pancreatic duct dilatation, biliary dilatation, and the encasement of peripancreatic vessels [2,46].

### 3.1. Neuroendocrine Tumors

Neuroendocrine tumors of the pancreas represent a clinically heterogeneous group of lesions, which account for 1–2% of all pancreatic neoplasms [47]. Most commonly, NETs are sporadic, but they may develop in association with familiar syndromes such as neurofibromatosis 1, Von Hippel–Lindau syndrome, tuberous sclerosis, or Wermer syndrome [47]. NETs are classified into functioning and non-functioning lesions. Due to hormone secretion, functioning tumors are suspected clinically early in the course of disease, and the role of imaging is to provide the precise localization of the tumor [47]. On the other hand, non-functioning tumors are commonly discovered late in the course of the disease and they present as a large lesions. Concerning differential diagnosis with solid SPNs, small NETs might pose a diagnostic dilemma. Similarly to SPNs, NETs commonly develop in the younger population [48]. On an MRI, small NETs are usually hypointense on T1-weighted images and display moderate to high signal intensity on T2-weighhted images, comparable to SPNs [47]. Furthermore, both entities display hyperintensity on diffusion-weighted images [20,48]. In such cases, the vascular profile might provide clues to a differential diagnosis. Namely, SPNs are slightly vascularized in the arterial phase, with progressive enhancement in the delayed phases, which is an uncommon pattern in NET [21]. In contrast, NETs are typically hypervascular in arterial phase images, enhancing to a higher extent than background pancreatic parenchyma [48] (Figure 5). They usually remain well enhanced throughout all postcontrast phases. According to the results of Yu et al., the most important differential features allowing distinction among small SPNs and NETs are the enhancement pattern and T2-weighted signal intensity, since all SPNs had an early heterogeneous and progressive enhancement in contrast to NETs, which showed intense, early, and persistent delayed enhancement [39]. Nevertheless, the hypovascular behavior of NETs, which is not infrequent among non-functioning tumors, might lead to preoperative misdiagnosis. In this setting, these tumors are hypointense in arterial phase imaging, with progressive enhancement on delayed phases comparable to SPNs [48]. In such cases, epidemiological data might be helpful, but a confidential differential diagnosis is sometimes impossible.

### 3.2. Pancreatic Adenocarcinoma

Pancreatic adenocarcinoma is the most common pancreatic tumor accounting for almost 90% of all malignant pancreatic neoplasms [49]. The incidence is highest in the seventh and eighth decade of life, with a male to female ratio of 2:1 [50]. Since the majority of patients have an advanced tumor stage at the time of clinical presentation, the diagnosis of PDAC is usually straightforward. The typical imaging presentation of irregular, solid mass (around 3 cm in diameter) with a propensity for local infiltration, including vascular encasement [51], favors the diagnosis of PDAC. Although PDACs are a heterogeneous group of tumors, the areas of internal necrosis or hemorrhage are very rare [51]. Non-contrast fat-suppressed T1-weighted images are useful for detection, since PDACs present as hypointense lesions in contrast to the normal background parenchyma, which has a high signal intensity [51]. The best delineation is possible on arterial phase imaging, where PDACs enhance poorly in contrast to highly enhanced background parenchyma, which can be explained by rich fibrous stroma inside the lesion and scarce tumor vascularization [52]. Due to the high volume of extracellular space, a PDAC is usually nearly isointense with the rest of the pancreatic parenchyma on the interstitial phase. The exceptions are large tumors, which commonly remain hypointense on delayed postcontrast phases. If all the above mentioned imaging features are present in combination with main pancreatic duct dilatation, the diagnosis of pancreatic ductal adenocarcinoma is straightforward. Nevertheless, small PDACs may lack all typical MRI features leading to misdiagnosis with other solid pancreatic lesions including small solid SPNs [39,44,49,52]. Epidemiological data might be useful for differential diagnosis as SPNs occur in younger women, while PDACs usually develop in the older population [50]. Nevertheless, PDAC might occur in younger patients, and SPNs may also be found in females in their third and fourth decades of life, and, although rarely, they may develop in males [2]. Concerning differential diagnosis between solid SPNs and small PDACs, T2-weighted image appearances might be helpful. SPNs are typically moderately to highly hyperintense on T2-weighted images, while PDACs are slightly hypointense or more frequently isointense with the rest of pancreatic parenchyma [22,39] (Figure 6). On T1-weighted images, both lesions are hypointense. Similarly to T1-weighted images, DWI does not provide additional information in differential diagnosis, since both SPNs and PDACs display restricted diffusion [39]. Distinct margins favor the diagnosis of SPNs, as PDAC is commonly ill-defined [38,44]. Concerning vascularity, the enhancement pattern of SPNs, which is described as gradual, is not that useful, as PDACs might also show progressive enhancement due to rich fibrotic stroma. 

The dilatation of the main pancreatic duct is the main differential feature, which suggests the diagnosis of PDAC, as it is rarely seen in SPN [20]. The approximated incidence of pancreatic duct dilatation in patients with SPN is about 10% [53]. The rarity of main pancreatic duct obstruction might be explained by the softness of SPNs. Although direct tumor infiltration of the pancreatic duct may be the cause of dilatation in patients with SPNs, in the majority of cases, only ductal compression was found on the histological analysis [6,54,55]. Additionally, the growth rate has also been pointed out as one potential factor for the main pancreatic duct dilatation, since in many large SPNs located in the pancreatic head, ductal dilatation does not occur [54]. In contrast, main pancreatic duct dilatation is one of the most important indirect signs of PDACs, as it is can be the only finding in very small lesions [52]. In addition, other secondary signs such as biliary dilatation and atrophy of the pancreatic parenchyma proximal to the lesion, are also very helpful for differential diagnosis, as they are frequently present in PDACs and seldom in SPNs [52]. From a clinical point of view, the correct preoperative differentiation between these entities is very important, since fewer invasive surgical procedures can be performed if SPNs are suspected, while more radical surgical approaches such as cephalic pancreaticoduodenectomy or distal pancreatectomy are necessary for PDACs.

### 3.3. Intrapancreatic Accessory Spleen

Intrapancreatic accessory spleen is a benign congenital anomaly with the presence of splenunculus inside the pancreatic parenchyma [44]. These lesions are commonly less than 2 cm in diameter and are located within 3 cm from the tip of pancreatic tail [56]. The MRI features, which favor the diagnosis of intrapancreatic accessory spleen, are well-defined margins, and the signal intensity is identical to that of the spleen in all MRI sequences [57]. Namely, intrapancreatic accessory spleen has a lower signal intensity on T1-weighted images, and a higher signal intensity on T2-weighted images in comparison to the rest of pancreatic parenchyma. If splenuncules are larger than 2 cm, a serpiginous postcontrast opacification on arterial phase images that is typical for spleen might be observed [58]. Nevertheless, if the characteristic enhancement pattern is not well seen due to the small size of the splenuncule, this benign entity might mimic other solid pancreatic lesions (Figure 7). Similarly to intrapancreatic accessory spleen, small SPNs display a high signal intensity on T2-weighted images, and restricted diffusion, which has been pointed out as important imaging features indicating splenuncule [57,58]. In difficult cases, a thorough comparison of the signal intensity of the lesion and splenic tissue must be performed. If there are still doubts, close monitoring is suggested or further diagnostic workup with Technetium 99 m sulfur colloid scintigraphy [58]. 

### 3.4. Pancreatic Metastases

Pancreatic metastases are rare, representing 2–5% of all malignant pancreatic neoplasms [59]. Renal cell carcinoma (RCC) and lung cancer are the most common primary tumors, which metastasize in the pancreas [59]. Less frequently they occur in breast and colon cancer, and malignant melanoma may also metastasize in the pancreas. Although the time interval between the diagnosis of the primary tumor and detection of the pancreatic metastases is usually less than 3 years, metastases from RCC might develop even 20 years after nephrectomy [59]. In the majority of patients, pancreatic metastases present as a solitary lesion (50–70% of cases), multifocal in 10–15%, and diffused in 15% up to 44% of cases [59,60]. On an MRI, metastases are hypointense on T1-weighted images, and are usually slightly hyperintense on T2-weighted images with restricted diffusion [60]. Vascular behavior depends on the angiogenic profile of primary tumor, with RCC metastases being typically hypervascular, while the rest of the metastatic lesions are usually hypovascular [60] (Figure 8). Small lesions (less than 2 cm) display homogeneous enhancement, whereas larger lesions show ring enhancement with a central hypoenhancing part due to poor perfusion [61]. Similarly to SPNs, the main pancreatic duct dilatation in pancreatic metastases is uncommon [62]. Taking into account only the imaging appearance, small SPNs might mimic solitary pancreatic metastasis. However, larger SPNs might rarely resemble centrally necrotic metastases. Nevertheless, in rare cases where there is a diagnostic dilemma, the patient’s medical history of a treated primary tumor enables an accurate diagnosis. 

## 4. Solid Pseudopapillary Neoplasm Mimicking Cystic Pancreatic Lesions

Large SPNs typically present as mixed solid and cystic lesions, mostly with solid parts at the periphery and heterogeneous cystic component located centrally [20]. The cystic component develops as a consequence of degeneration due to insufficient blood supply. Probably the most pathognomonic finding of large SPNs is internal hemorrhage as it is rarely present in other pancreatic cystic tumors [2,6]. Due to the hemorrhagic degeneration, a fluid–fluid level might be seen, which is also infrequent in other cystic pancreatic lesions [20,63]. Nevertheless, hemorrhage may be absent or hardly visible, thus making the differential diagnosis of SPNs with other cystic lesions difficult. Additionally, a cystic component of SPNs might rarely be organized in a multilocular pattern due to the development of thin internal septations. Furthermore, a purely cystic SPN has also been described [48]. In such cases, distinction from other cystic pancreatic tumors is challenging [63,64].

### 4.1. Serous Cystadenoma

Serous cystadenomas are benign cystic pancreatic tumors that usually occur in older women [65]. SCN consists of numerous small cysts (less than 20 mm) arranged in a “honeycomb pattern”. The cysts are separated by fibrous septa, which radiate from a central scar. The presence of coarse calcifications in the central scar is considered typical for SCN [65]. The contour of SCN is characteristically lobulated corresponding to the “cyst on cyst” internal structure. On an MRI, T2-weighted images clearly depict the microcystic nature of SCNs with a hyperintense internal content [65,66]. On T1-weighted images, the lesion is hypointense without hyperintense foci. After intravenous contrast administration, the septa enhances, while the central scar remains hypointense [65]. Both SPNs and SCNs are well-defined, and rarely accompanied by main pancreatic duct dilatation. In addition, similarly to SPNs, SCNs are commonly detected incidentally, without clinical symptomatology. However, if typical imaging features are present, differential diagnosis between these lesions is not difficult. Nevertheless, SCN might have an atypical imaging appearance with an internal hemorrhagic content (Figure 9) or it can present in the form of an extremely microcystic lesion mimicking a solid pancreatic tumor [67] (Figure 10). Since in such cases the lesion consists of numerous tiny, microscopic serous cysts, an avid enhancement of their walls may masquerade a solid tumor.

Mixed micro- and macrocystic SCNs may resemble a solid tumor accompanied by cystic degeneration, such as SPNs [68]. In this setting, T2-weghted images and MRCP may solve diagnostic dilemma, demonstrating the microcystic nature of SCN (Figure 11). Very rarely, an SPN might present as a lobulated lesion with central calcifications, similarly to an SCN. Nevertheless, the appearance on T2-weighted images should allow preoperative distinction as SCN typically present as hyperintense lesions, indicating the cystic nature of the tumor, while SPNs display a solid or mixed solid-cystic appearance [68]. 

### 4.2. Mucinous Cystic Neoplasm

Mucinous cystic neoplasm is rare cystic neoplasm, which similarly to SPNs, has a clear female predilection, and usually appears as a round tumor most commonly located in the body or tail of the pancreas [69]. MCN usually presents as a unilocular or mildly septate round cystic lesion with a thick wall, which enhances in delayed postcontrast MR images. Since the cyst wall is lined by mucin-secreting epithelial cells, a protein-rich internal content may display T1-weighted hyperintensity leading to confusion with hemorrhage, which is typical for SPN [68]. Taking into account the fact that both lesions are round and cystic, with heterogeneous internal content, and found in young females, a differential diagnosis between MCN and SPN might be challenging. However, the useful tip for distinction among these tumors is the presence of solid component, which favors the diagnosis of SPNs [68] (Figure 12). While MCNs are originally cystic tumors, SPNs grow initially as solid lesions, which undergo cystic degeneration leading to a very heterogeneous MRI appearance. Additionally, the thick inner septa and outer cyst wall are characteristic for MCNs [68]. Moreover, while SPENs typically occur in young females, MCNs occur in middle-aged women [69]. If SPN presents as an entirely cystic lesion, or if a solid component is present in MCN, preoperative differentiation between these tumors might not be possible.

### 4.3. Walled-Off Necrosis

One of the complications of acute necrotic pancreatitis is the development of fluid collections in pancreatic parenchyma, which are termed walled-of necrosis (WON) [70]. WON is the most common cystic lesion of the pancreas [70]. Since the internal content of these collections contains necrotic tissues and frequently hemorrhage, these collections present on an MRI as heterogeneous lesions with a varying signal intensity on both T1-weighted and T2-weighted images [71]. Commonly, there are areas of T1-weighted hyperintensity due to blood products and necrotic or proteinaceous debris [71]. These cystic lesions are well limited with a thickened wall, which opacifies on postcontrast images corresponding to fibrosis and granulation tissue. Peripheral calcifications might also be present [71]. Furthermore, some internal septations may exist. On the basis of the above mentioned imaging features, it is obvious that WON may mimic SPN. MRI findings, which favor WON, include the absence of the solid soft-tissue component, and the absence of postcontrast opacification [71] (Figure 13). Due to the high signal intensity on native T1-weighted images, it might be difficult sometimes to assess postcontrast enhancement. In such cases, subtraction images are useful. 

Nevertheless, if WON is small, postcontrast opacification of the pseudocapsule and surrounding pancreatic parenchyma may simulate peripheral opacification in SPNs [72] (Figure 14). In the appropriate clinical context, including the presence of stigmata of acute or chronic pancreatitis or a medical history with previous attacks of severe abdominal pain, imaging features are reasonably suggestive for the correct diagnosis of WON [72]. Differential diagnosis is difficult if there are no clinical data or imaging findings of previous attacks of acute or chronic pancreatitis. In this regard, a short-term follow-up is helpful, since the MR imaging appearance of WON evolves over time, whereas SPNs exhibit no significant size change [72].

### 4.4. Cystic Neuroendocrine Tumors

Large neuroendocrine tumors, especially those that are non-functioning, might undergo cystic degeneration, thus presenting as mixed solid and cystic neoplasms in 10% of cases [73,74]. Contrary to cystic changes that occur due to infarction and liquefaction necrosis, the cystic component in NET is not the only consequence of necrosis. Namely, the cysts are often lined by well-preserved neoplastic endocrine cells with their cavities filled with clear fluid, thus indicating the different pathophysiology of cystic NET [75]. There is also a hypothesis that bleeding in these highly vascular lesions may be the initial event preceding cystic formation [76]. The cystic component might be unilocular and located centrally, but it may also show micro- or macrocystic pattern [74]. When NETs present as large heterogeneous lesions consisting of both solid and cystic part, misdiagnosis with SPNs is possible (Figure 15). 

However, in contrast to SPNs, the cystic component in NETs is generally not hemorrhagic, while intralesional hemorrhage is a pathognomonic finding in SPNs [76]. Nevertheless, if hemorrhagic areas are also present in NETs, the differential diagnosis might be challenging (Figure 16). Furthermore, calcification, either peripheral or central, might be present in both types of lesions [73,75]. In addition, the clinical presentation is usually very similar because, like SPNs, non-functioning NETs are usually detected when they reach a large size. The most important differential imaging feature, which allows the distinction between large cystic NETs and SPNs, is the postcontrast enhancement pattern [73]. Namely, the presence of a diffused, intense, heterogeneous enhancement of the solid component in the early arterial phase favors the diagnosis of NET [73]. Commonly, NETs show a higher degree of enhancement in comparison to normal pancreatic parenchyma [73]. On the other hand, SPNs are less enhanced than normal parenchyma, and they display a slight heterogeneous peripheral enhancement on the early arterial phase with progressive fill-in on the delayed phase [20].

## 5. Conclusions

In conclusion, we have provided a detailed review of the imaging features of SPNs. Most commonly, SPNs present as well-demarcated, encapsulated lesions, consisting of both cystic and solid components, which are present in variable proportions. The characteristic imaging feature is the internal hemorrhage, which can be seen as areas of T1-weighted hyperintensity. The enhancement of the solid component is lower in comparison to the rest of the pancreas in the arterial phase, with progressive enhancement in the portal-venous and delayed phase. The impact of pathological characteristics on an MRI presentation was also pointed out. In addition, we presented a systematic overview of solid and cystic pancreatic lesions, which may mimic SPNs. The analysis of lesion appearances on T1-weighted, T2-weighted images, DWI, and postcontrast enhancement patterns, in combination with clinical and epidemiological data, are necessary for accurate lesion characterization. Taking into account different treatment strategies for SPNs and lesions simulating their appearance on imaging, it is clinically very important to preoperatively differentiate SPNs from their mimickers in order to provide optimal treatment for each patient. 

## Figures and Tables

**Figure 1 diagnostics-13-01074-f001:**
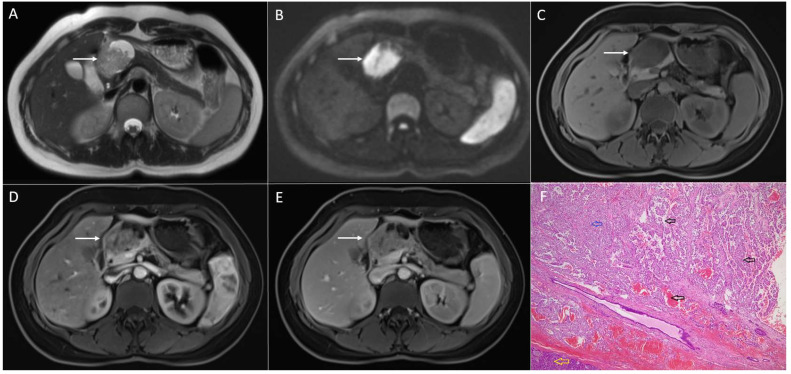
Solid pseudopapillary neoplasm of the pancreas in 25-year-old woman. On axial T2-weighted image (**A**), a mixed solid and cystic tumor (*arrow*) located in the pancreatic head is seen. The solid part of the lesion has high signal intensity on DWI (*arrow*) (**B**), intermediate signal intensity on T1-weighted image (**C**), with heterogeneous enhancement in arterial phase (**D**) and progressive enhancement in portal-venous phase (**E**). Hematoxylin and eosin (H&E) staining (**F**) showed pseudopapillary structures (*black arrows*), solid part of the tumor (*blue arrow*), and normal pancreatic parenchyma next to the tumor (*yellow arrow*); original magnification ×400.

**Figure 2 diagnostics-13-01074-f002:**
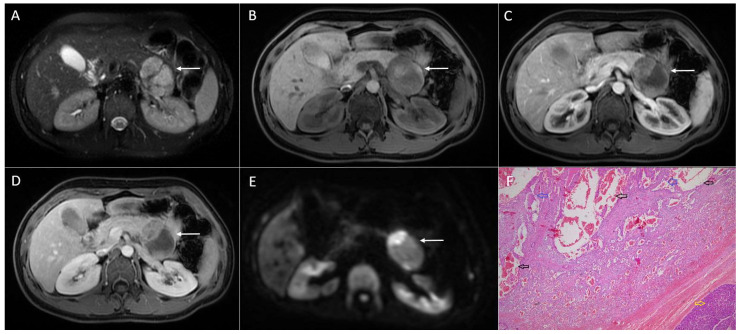
Solid pseudopapillary neoplasm of the pancreas in 33-year-old woman. Well-demarcated heterogeneously hyperintense lesion (*arrow*) on T2-weighted FS image (**A**) located in the pancreatic tail is shown. On native T1-weighted FS image (**B**), the lesion (*arrow*) is mainly hypointense with small hyperintense areas indicating hemorrhage, showing only discrete enhancement of the solid component in arterial phase (**C**), and slow progressive enhancement in portal-venous phase (**D**). On DWI, solid part of the lesion shows restricted diffusion (**E**). Hematoxylin and eosin (H&E) staining (**F**) showed microcystic areas (*black arrows*), with pseudopapillary structures inside of them (*blue arrows*), and normal pancreatic parenchyma next to the tumor (*yellow arrow*); original magnification ×400.

**Figure 3 diagnostics-13-01074-f003:**
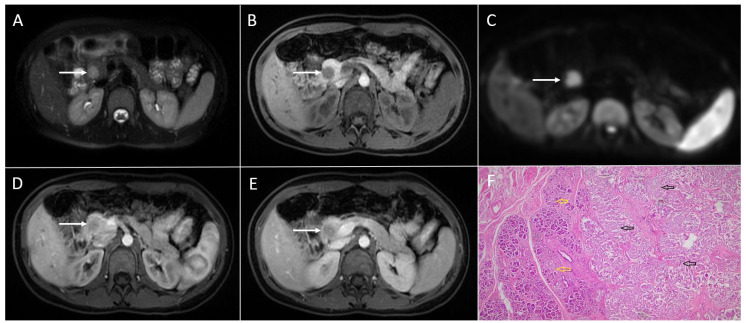
Solid pseudopapillary neoplasm of the pancreas in 18-year-old woman. Axial T2-weighted FS image (**A**) shows slightly round hyperintense lesion in the pancreatic head (*arrow*). The tumor is hypointense on T1-weighted image (**B**) with restricted diffusion on DWI (b = 800 s/mm^2^) (**C**). In arterial phase, the tumor shows discrete enhancement (**D**) and remains hypointense in portal-venous phase (**E**). Hematoxylin and eosin (H&E) staining (**F**) showed solid pseudopapillary tumor (*black arrow*). Normal pancreatic parenchyma is also shown (*yellow arrow*); original magnification ×400.

**Figure 4 diagnostics-13-01074-f004:**
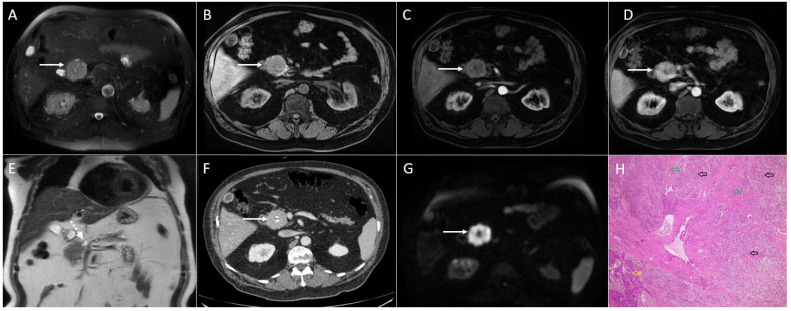
Sixty-eight-year-old man with solid pseudopapillary tumor. Axial T2-weighted FS image (**A**) shows slightly lobulated hyperintense lesion in the pancreatic head (*arrow*). On native T1-weighted FS image (**B**), the tumor is hypointense. After intravenous contrast administration, the lesion is only slightly enhanced in arterial phase (**C**). Portal-venous phase in the same patient (**D**) shows progressive enhancement of the lesion. The tumor leads to the obstruction of the main pancreatic duct, which is dilatated in the body and the tail of the pancreas as is seen on coronal T2-weighted image (*dashed*
*arrow*) (**E**). On the axial DWI (**F**), the tumor shows restricted diffusion with central hypointense area (*arrow*). Corresponding CT image (**G**) better depicts central scar with punctiform calcifications (*arrow*). Hematoxylin and eosin (H&E) staining (**H**) showed solid pseudopapillary tumor (*black arrows*) with fibrous septations (*green arrows*). Normal pancreatic parenchyma is also shown (*yellow arrow*); original magnification ×400.

**Figure 5 diagnostics-13-01074-f005:**
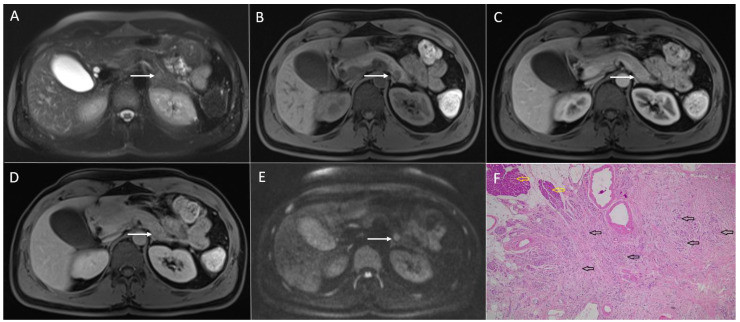
Thirty-five-year-old man with neuroendocrine tumor. On T2-weighted FS image (**A**), the tumor is hardly visible as slight hyperintense lesion in the posterior aspect of the pancreatic body (*arrow*). The tumor is clearly seen on T1-weighted native FS image (**B**), as hypointense lesion (*arrow*). In arterial (**C**) and portal-venous phase (**D**), the tumor is not visible as it enhances to the same degree as the surrounding pancreatic parenchyma. The lesion displays high signal intensity on DWI (**E**). Hematoxylin and eosin (H&E) staining (**F**) showed neuroendocrine tumor G2 with marked desmoplastic reaction (*black arrows*). Normal pancreatic parenchyma is also shown (*yellow arrows*); original magnification ×400.

**Figure 6 diagnostics-13-01074-f006:**
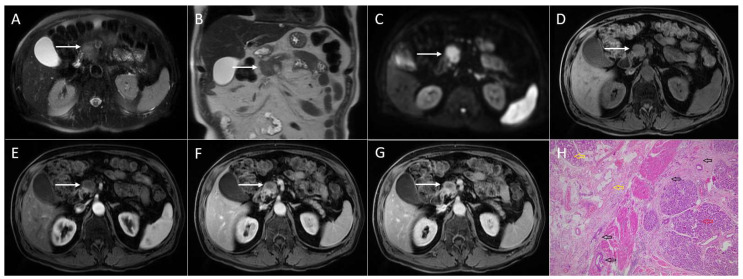
Pancreatic adenocarcinoma in 43-year-old women. Axial T2-weighted FS image (**A**) shows moderately hyperintense tumor (*arrow*) located in the pancreatic head. On coronal T2-weighted image (**B**), no obstruction of the main pancreatic duct is seen. The lesion (*arrow*) is hyperintense on DWI (**C**). On T1-weighted fat-suppressed image (**D**), the tumor is hypointense (*arrow*) and hypovascular in arterial phase (**E**), with discrete progressive enhancement in portal-venous phase (**F**) and delayed phase (**G**). Hematoxylin and eosin (H&E) staining (**H**) showed well-differentiated ductal adenocarcinoma (*black arrows*), which infiltrates pancreatic parenchyma (*red arrow*) and duodenal wall (*yellow arrows*); original magnification ×400.

**Figure 7 diagnostics-13-01074-f007:**
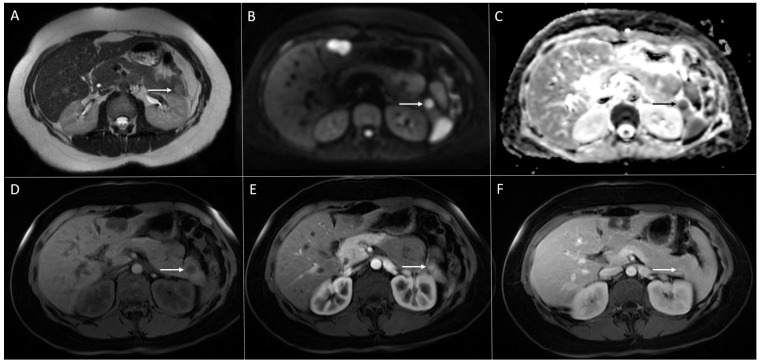
Intrapancreatic accessory spleen in 56-year-old women. Axial T2-weighted image (**A**) shows small slightly round hyperintense lesion (*arrow*) located in the pancreatic tail. On DWI (**B**), the lesion is hyperintense with low signal intensity on corresponding ADC map (**C**). On T1-weighted fat-suppressed image (**D**), the lesion (*arrow*) is hypointense with intense enhancement in arterial phase (**E**) remaining well-vascularized on portal-venous phase (**F**).

**Figure 8 diagnostics-13-01074-f008:**
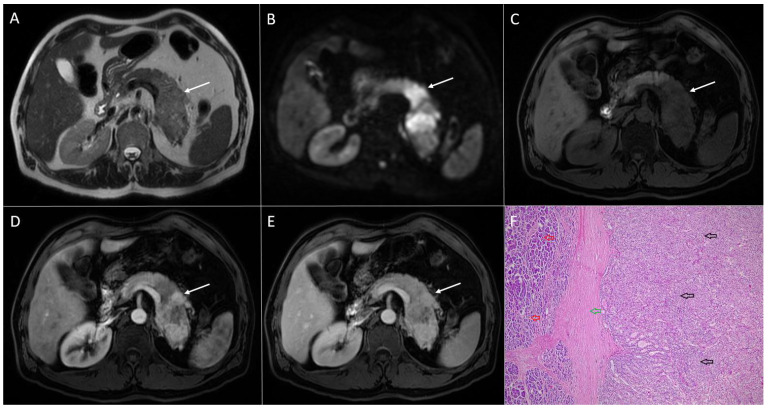
Pancreatic metastasis in 62-year-old man. T2-weighted image (**A**) shows slightly heterogeneous hyperintense lesion (*arrow*) in the pancreatic tail. On DWI (**B**), the tumor displays restricted diffusion. On T1-weighted FS image (**C**), the lesion (*arrow*) is hypointense. The lesion is well-vascularized in arterial (**D**) with persistent enhancement in portal-venous phase (**E**). Hematoxylin and eosin (H&E) staining (**F**) showed clear cell renal carcinoma metastasis (*black arrows*) and normal pancreatic parenchyma (*red arrows*) separated by fibrous pseudocapsula (*green arrow*) from the tumor; original magnification ×400.

**Figure 9 diagnostics-13-01074-f009:**
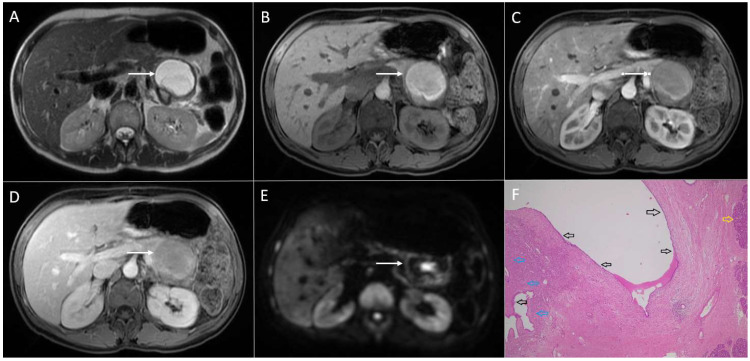
Non-mucinous cystadenoma of the pancreas in 42-year-old woman. T2-weighted image (**A**) shows septated cystic lesion (*arrow*) in the pancreatic tail. On T1-weighted FS image (**B**), the lesion (*arrow*) is hyperintense indicating hemorrhagic content. The lesion remains hyperintense in arterial (**C**) and portal-venous phase (**D**) due to high inner signal intensity on native T1-weghted image but without detectable vascularization on subtracted images (not shown). On DWI (**E**), the tumor displays central area of restricted diffusion. Hematoxylin and eosin (H&E) staining (**F**) showed non-mucinous cystadenoma lined with bilio-pancreatic epithelium (*black arrows*), and ovarian-like stroma in the wall of the cyst (*blue arrows*). Normal pancreatic parenchyma is also seen (*yellow arrow*); original magnification ×400.

**Figure 10 diagnostics-13-01074-f010:**
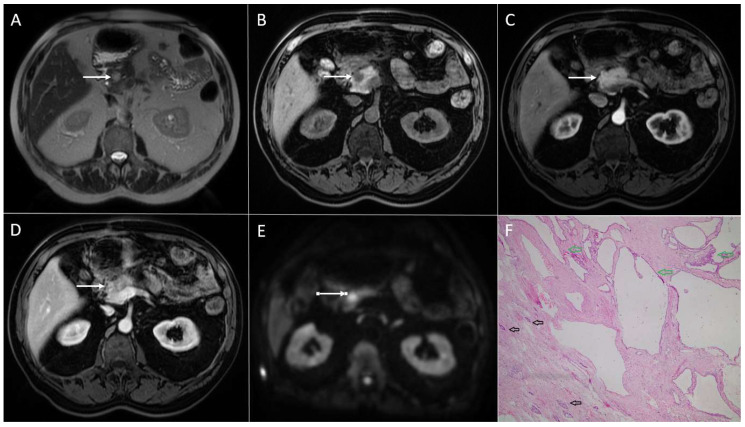
Serous cystadenoma in 55-year-old men. On T2-weighted image (**A**), a small round hyperintense lesion (*arrow*) is seen in the pancreatic head. The lesion is hypointense on T1-weighted fat-suppressed image (**B**), and is well-vascularized in arterial phase (**C**) with persistent enhancement in portal-venous phase (**D**). On diffusion-weighted image the lesion, (*arrow*) shows high signal intensity (**E**). Hematoxylin and eosin (H&E) staining (**F**) showed multiple microcystic spaces (*green arrows*) divided by fibro-sclerotic septas. Normal pancreatic parenchyma is also seen (*black arrows*); original magnification ×400.

**Figure 11 diagnostics-13-01074-f011:**
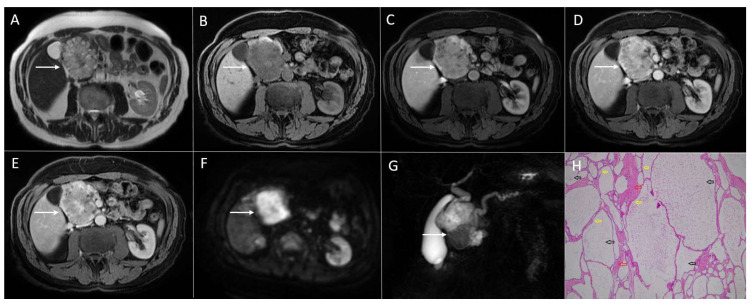
Serous cystadenoma in 68-year-old men. On T2-weighted image (**A**), a lobulated hyperintense lesion (*arrow*) is seen in the pancreatic head. Note fibrous scar in the center of the lesion. The lesion (*arrow*) is hypointense on T1-weighted image (**B**) with slight enhancement in arterial phase (**C**) and progressive enhancement in portal-venous (**D**) and delayed phase (**E**). On DWI (**F**), the lesion has high signal intensity. MRCP (**G**) nicely depicts microcystic nature of the lesion. Hematoxylin and eosin (H&E) staining (**H**) showed multiple microcystic spaces lined with single row cubic epithelium (*black arrows*), and small uniform nuclei without atypical features (*yellow arrows*). Fibro-sclerotic septas between small cysts are also shown (*red arrows*); original magnification ×400.

**Figure 12 diagnostics-13-01074-f012:**
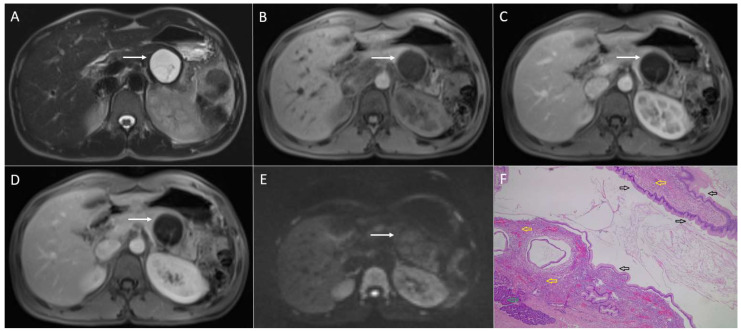
Mucinous cystadenoma in 39-year-old woman. On T2-weighted image (**A**) of a well-demarcated cystic lesion (*arrow*) with discrete internal septations is seen in the pancreatic tail. On T1-weighted fat-suppressed image (**B**), the lesion is hypointense (*arrow*) without enhancement in arterial phase except in septations (**C**), which is better depicted in portal-venous (**D**) phase. The lesion (*arrow*) does not show diffusion restriction (**E**). Hematoxylin and eosin (H&E) staining (**F**) showed mucinous cystadenoma with mucinous epithelium lining cyst (*black arrows*). Ovarian stroma is also depicted (*yellow arrows*). Normal pancreatic parenchyma is seen next to the tumor (*green arrow*); original magnification ×400.

**Figure 13 diagnostics-13-01074-f013:**
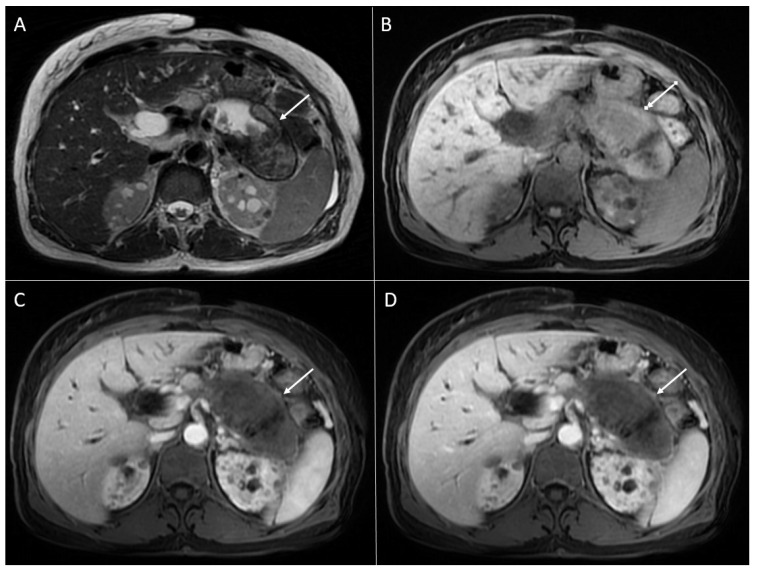
Walled-of necrosis in 54-year-old woman. On T2-weighted image (**A**), large heterogeneous partly cystic lesion (*arrow*) is seen in the pancreatic body and the tail. The lesion (*arrow*) is predominantly hyperintense on T1-weighted FS image (**B**) without opacification in late arterial phase (**C**) and portal-venous phase (**D**).

**Figure 14 diagnostics-13-01074-f014:**
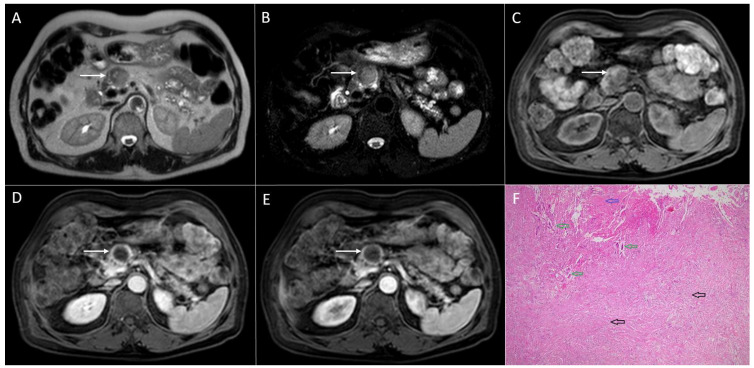
Walled-of necrosis in 44-year-old man. A round lesion (*arrow*) slightly hyperintense on both T2-weighted image (**A**) and T2-weighted fat-suppressed image (**B**) is seen in pancreatic head. The lesion (*arrow*) is hypointense on T1-weighted image (**C**) without enhancement in arterial phase (**D**) and portal-venous (**E**). Note only enhancing wall of the lesion. Hematoxylin and eosin (H&E) staining (**F**) showed walled-of necrosis with fibro-hyalinized wall of the cyst (*black arrows*), numerous histiocytes and foreign body giant cells (*green arrows*). Necrotic debris is also shown (*blue arrow*); original magnification ×400.

**Figure 15 diagnostics-13-01074-f015:**
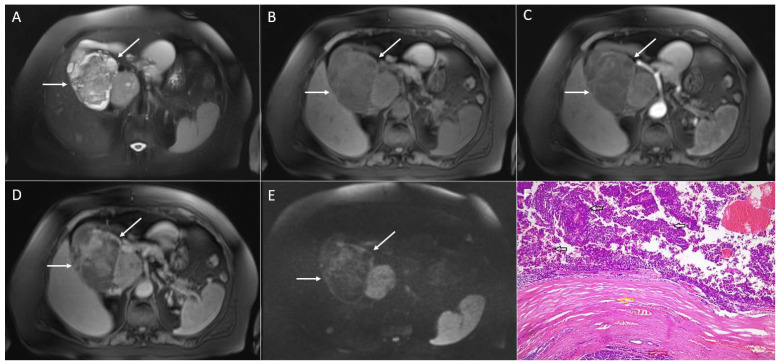
Neuroendocrine tumor in 55-year-old woman. Mixed solid and cystic bilobar lesion is seen in pancreatic head with heterogeneous signal intensity on T2-weighted FS image (**A**). The lesion is hypointense on T1-weighted FS image (**B**) with only slight enhancement in arterial phase (**C**) and progressive enhancement of solid portion in portal-venous phase (**D**). Solid part of the lesion displays high signal intensity on DWI (**E**). Hematoxylin and eosin (H&E) staining (**F**) showed neuroendocrine tumor (*black arrows*) with hemorrhagic foci, partly demarcated with fibro-hyalinized capsule (*yellow arrow)* from normal pancreatic parenchyma (*red arrow*); original magnification ×400.

**Figure 16 diagnostics-13-01074-f016:**
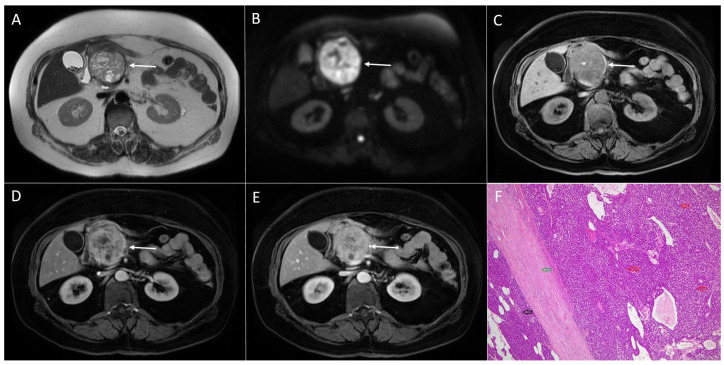
Neuroendocrine tumor in 43-year-old woman. Axial T2-weighted image (**A**) shows round heterogeneous lesion with internal hyperintense areas (*arrow*) located in the pancreatic head. On DWI, the tumor displays high signal intensity (**B**). The lesion (*arrow*) is hypointense on native T1-weighted FS image with small hyperintense focus centrally located (**C**). In arterial phase (**D**), the tumor shows intense heterogeneous enhancement, which persists in portal-venous phase (**E**). Hematoxylin and eosin (H&E) staining (**F**) showed neuroendocrine tumor G3 (*red arrows*). Normal pancreatic parenchyma (*black arrow*) is also seen separated from the tumor by thick fibrous capsule (*green arrow*); original magnification ×400.

## Data Availability

Not applicable.

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
