# Peer review of "The Role of MRI in the Diagnosis of Solid Pseudopapillary Neoplasm of the Pancreas and Its Mimickers: A Case-Based Review with Emphasis on Differential Diagnosis"

_diagnostics, 2023, doi:10.3390/diagnostics13061074_

Round 1
Reviewer 1 Report
COMMENTS TO THE AUTHOR(S)
This is a review of ID: diagnostics-2254677, The role of MRI in the diagnosis of solid pseudopapillary neoplasm of the pancreas: a case-based review with emphasis on differential diagnosis”. This is a good quality manuscript with a detailed summary of imaging findings, but needs some minor revisions.
Major Comments:
Although the article is intended as a summary of MRI imaging findings in SPN, more than half of the content is a summary of imaging findings in other pancreatic tumors. If readers read the manuscript after seeing the title of the paper, it is likely that they will read through information they are not looking for in the middle of the manuscript. It would be better to add a description of other pancreatic tumors in the title.
Minor comments
1) Images
The images are not uniform in size. They should be uniform, enlarged, and presented more clearly. The image quality should allow detailed observation of tumor margins, boundaries, internal structure, signal intensity, etc.
2) EUS has an important role in the imaging of pancreatic tumors; since MRI or CT is not the only imaging modality, why not describe the role of MRI, including its role in relation to other modalities? This paper does not describe EUS as well. How about adding the role of MRI, which is useful in clinical practice, such as considering the indication for EUS-FNA based on MRI imaging findings, rather than diagnostic imaging alone?
Reviewer 2 Report
An interesting manuscript which needs a few minor amendments prior to resubmission:
1 - Please add information regarding ethics approval and criteria for patient selection
2 - English requires small correction (a few spelling and grammar mistakes were spotted)
3 - Please consider adding a discussion chapter and characterise all specific criteria for the SPN and unique parameters differentiating it from other lesions. A few paragraphs - for the reader to get the main idea of the manuscript in the end.
